# Surface Proteome of Extracellular Vesicles and Correlation Analysis Reveal Breast Cancer Biomarkers

**DOI:** 10.3390/cancers16030520

**Published:** 2024-01-25

**Authors:** Nico Hüttmann, Yingxi Li, Suttinee Poolsup, Emil Zaripov, Rochelle D’Mello, Vanessa Susevski, Zoran Minic, Maxim V. Berezovski

**Affiliations:** 1Department of Chemistry and Biomolecular Sciences, University of Ottawa, Ottawa, ON K1N 6N5, Canada; nhutt069@uottawa.ca (N.H.); yli840@uottawa.ca (Y.L.); spool093@uottawa.ca (S.P.); ezari068@uottawa.ca (E.Z.); rdmel035@uottawa.ca (R.D.); vsuse027@uottawa.ca (V.S.); 2John L. Holmes Mass Spectrometry Facility, Faculty of Science, University of Ottawa, Ottawa, ON K1N 6N5, Canada; zminic@uottawa.ca

**Keywords:** breast cancer, extracellular vesicles, surface proteins, proteomics, biomarkers

## Abstract

**Simple Summary:**

Breast cancer (BC) is the second leading cause of death in Canadian women. Despite this, non-invasive early detection methods for BC have yet to be developed. Cells release various types of extracellular vesicles (EVs) as intercellular signals into biofluids, such as blood, reflecting the physiological state of the origin cell. It is by transmitting bioactive molecules between cells that EVs may regulate multiple pathologies in human diseases, including cancer. The detection of protein biomarkers presented on the surface of EVs in biofluids could enable the early diagnosis of BC. Therefore, we studied the surfactome of EVs from cancerous and non-cancerous cells and found several potential biomarkers that may be involved in the early diagnosis of BC.

**Abstract:**

Breast cancer (BC) is the second most frequently diagnosed cancer and accounts for approximately 25% of new cancer cases in Canadian women. Using biomarkers as a less-invasive BC diagnostic method is currently under investigation but is not ready for practical application in clinical settings. During the last decade, extracellular vesicles (EVs) have emerged as a promising source of biomarkers because they contain cancer-derived proteins, RNAs, and metabolites. In this study, EV proteins from small EVs (sEVs) and medium EVs (mEVs) were isolated from BC MDA-MB-231 and MCF7 and non-cancerous breast epithelial MCF10A cell lines and then analyzed by two approaches: global proteomic analysis and enrichment of EV surface proteins by Sulfo-NHS-SS-Biotin labeling. From the first approach, proteomic profiling identified 2459 proteins, which were subjected to comparative analysis and correlation network analysis. Twelve potential biomarker proteins were identified based on cell line-specific expression and filtered by their predicted co-localization with known EV marker proteins, CD63, CD9, and CD81. This approach resulted in the identification of 11 proteins, four of which were further investigated by Western blot analysis. The presence of transmembrane serine protease matriptase (ST14), claudin-3 (CLDN3), and integrin alpha-7 (ITGA7) in each cell line was validated by Western blot, revealing that ST14 and CLDN3 may be further explored as potential EV biomarkers for BC. The surface labeling approach enriched proteins that were not identified using the first approach. Ten potential BC biomarkers (Glutathione S-transferase P1 (GSTP1), Elongation factor 2 (EEF2), DEAD/H box RNA helicase (DDX10), progesterone receptor (PGR), Ras-related C3 botulinum toxin substrate 2 (RAC2), Disintegrin and metalloproteinase domain-containing protein 10 (ADAM10), Aconitase 2 (ACO2), UTP20 small subunit processome component (UTP20), NEDD4 binding protein 2 (N4BP2), Programmed cell death 6 (PDCD6)) were selected from surface proteins commonly identified from MDA-MB-231 and MCF7, but not identified in MCF10A EVs. In total, 846 surface proteins were identified from the second approach, of which 11 were already known as BC markers. This study supports the proposition that Evs are a rich source of known and novel biomarkers that may be used for non-invasive detection of BC. Furthermore, the presented datasets could be further explored for the identification of potential biomarkers in BC.

## 1. Introduction

Extracellular vesicles (Evs) are a collective term for phospholipid bilayer-enclosed structures that are secreted by all cell types and species across the tree of life [1,2]. Evs play a role in communication between proximal or distant cells [3]. Currently, Evs are classified into three main subtypes based on their intracellular origin and secretion mechanism: exosomes, microvesicles (MVs), and apoptotic bodies [4]. Evs can also be classified based on their diameter. Small extracellular vesicles (sEVs) are approximately between 30 and 150 nm in diameter [5,6], while medium extracellular vesicles (mEVs) are on average larger and in the range of 100 nm to 1000 nm [1,7]. In this study, isolated vesicles with a median diameter in the range of 105–130 and 150–160 nm in diameter were referred to as sEVs and mEVs, respectively. Currently, Evs are understood to play a role in communication between cells by transferring the bioactive molecular cargoes of various molecules such as nucleic acids, proteins, and metabolites [3,8,9,10,11]. The content of EV cargo depends on the condition of cells and, in general, differs from healthy to diseased cells. Therefore, the exchange of bioactive molecules between the Evs in healthy and diseased cells can naturally play a role in tumor progression, metastasis, immunosuppression, and drug resistance [12,13,14,15].

The development of a diagnostic platform with the ability to detect a molecular signature of cancer progression may serve as a biomarker for early detection of disease [16]. Therefore, proteins, along with other biomolecules, compose the cargo of Evs, making them a promising source of biomarkers for the non-invasive detection of various diseases [17], given that they can be isolated from all biofluids [18]. MS-based proteomics enabled the discovery of protein markers that are widely used for confirming the presence of Evs: CD63, CD9, and CD81 [19]. Moreover, proteomics profiling of sEVs allowed subtyping of BC cell lines [20]. This investigation motivated further exploration of BC cell line-derived Evs to identify surface biomarkers that could be directly detected in biofluids for BC diagnosis by either antibody or aptamer-based diagnostic techniques [21]. Several studies have been extensively used to identify cell surface proteins on sEVs [22,23,24,25,26,27,28,29,30]. Recently, a novel workflow was developed for surface proteomics analysis of small (30–150 nm) and larger Evs (100–800 nm) derived from SW620, LIM1863, MDA-MB-231, and U87 cells [30]. This workflow is based on using membrane impermeant biotin to label and enrich surface proteins and identify them using mass spectrometry.

In this study, two proteomics-based approaches for the discovery of potential surface EV protein biomarkers for BC are demonstrated. Evs were isolated by differential ultracentrifugation (dUC) from three epithelial breast cell lines with varying degrees of malignancy: MDA-MB-231, MCF7, and non-malignant MCF10A cells. The first approach is based on cell line-specific identification of sEV proteins. Surface proteins were predicted by UniProt annotations of membrane proteins, and cluster analysis predicted co-localization of potential biomarkers with EV marker proteins. The presence of selected markers is then validated by WB analysis to assess their potential as BC markers. The second approach uses a biotinylation reagent to experimentally identify proteins that are accessible from the surface of Evs. Proteins common to Evs from MDA-MB-231 and MCF7 but not identified in MCF10A Evs were considered cancer EV markers. These proteins were then annotated with disease terms to identify known BC-associated proteins that can be detected from biofluids on the surface of Evs.

## 2. Materials and Methods

### 2.1. Cell Lines and Cell Cuture

MDA-MB-231 (ATCC^®^ HTB-26™), MCF7 (ATCC^®^ HTB-22™) and MCF10A (ATCC^®^ CRL-10317™) cell lines were obtained from ATCC (Cedarlane Corporation, Burlington, ON, Canada). MDA-MB-231 and MCF7 were cultured in DMEM/F12 growth medium (Gibco™, Thermo Fisher Scientific, Mississauga, ON, Canada), supplemented with 10% (*v*/*v*) fetal bovine serum (FBS) (Cat No. F2442, MilliporeSigma, Oakville, ON, Canada), 100 U/mL penicillin, and 100 µg/mL streptomycin (Cat No. A5955, MilliporeSigma, Oakville, ON, Canada). MCF10A was cultured in DMEM/F12 growth medium supplemented with 5% (*v*/*v*) horse serum (HS) (Cat No. H1270, MilliporeSigma, Oakville, ON, Canada), 100 U/mL penicillin and 100 µg/mL streptomycin, 20 ng/mL epidermal growth factor (EGF) (Cat No. GMP100-15, Peprotech, Mississauga, ON, Canada), 0.5 mg/mL hydrocortisone (Cat No. H-0888, MilliporeSigma, Oakville, ON, Canada), 10 µg/mL insulin (Cat No. I-1882, MilliporeSigma, Oakville, ON, Canada), and 100 ng/mL cholera toxin (Cat No. C-8052, MilliporeSigma, Oakville, ON, Canada) [31]. FBS and HS were diluted 1:1 with a growth medium and centrifuged for 20 h at 100,000× *g* to prepare EV-depleted growth medium. All cell lines were grown in EV-depleted cell medium in a humidified incubator at 37 °C with 5% CO_2_ for 48 to 72 h until they reached ~90% confluence prior to the collection of cell culture supernatant.

### 2.2. EV Isolation by Differential Ultracentrifugation (dUC)

The EV isolation protocol for small and large EVs was adapted from previous protocols by calculating the k-factor, a measure of pelleting efficiency [32]. Approximately 37 mL of EV-containing cell culture supernatant was collected from each cell culture dish and transferred to Falcon tubes separately. Cells were depleted by centrifugation at 300× *g* for 10 min at 4 °C. The supernatant was decanted into another Falcon tube, and cell debris and larger vesicles, such as apoptotic bodies, were depleted at 2000 *g* for 20 min at 4 °C. Frozen samples were thawed overnight at 4 °C and centrifuged again at 2000× *g* for 20 min at 4 °C before the supernatant was transferred to ultracentrifuge tubes. The following centrifugation steps were carried out using a Beckman Coulter XL-A Analytical Ultracentrifuge with ultracentrifuge tubes (Polypropylene, Open-Top Thinwall centrifuge tubes, 38.5 mL) and a SW 28 swinging bucket-rotor (all Beckman Coulter, Mississauga, ON, Canada). mEVs were isolated for 1 h at 4 °C at 16,500 *g* (k-factor: 1541, 11,200 rpm). The supernatant was collected for sEV isolation for 3 h at 4 °C at 100,000 *g* (k-factor: 256, 27,600 rpm). Co-isolated contaminant proteins were removed from mEVs and sEVs by resuspension in phosphate-buffered saline (PBS) (Gibco™, Thermo Fisher Scientific, Mississauga, ON, Canada), pH 7.4, and an additional centrifugation step at respective conditions. Pellets were finally resuspended in 100 µL of PBS and frozen at −20 °C until further processing.

### 2.3. Determination of Protein Concentration

The protein content of individual samples was measured with the Bradford assay. Approximately 4 µL of solubilized EV samples were incubated with 10 µL of Coomassie reagent (Pierce™ Coomassie Plus Bradford, Thermo Fisher Scientific, Mississauga, ON, Canada) for 10 min at room temperature (RT). The absorption was read at a wavelength of 595 nm using a NanoDrop™ One Microvolume UV-Vis Spectrophotometer (Thermo Fisher Scientific, Mississauga, ON, Canada). The calibration curve was prepared with bovine serum albumin (BSA) (MilliporeSigma Canada Ltd., Oakville, ON, Canada).

### 2.4. Nanoparticle Tracking Analysis (NTA)

A ZetaView nanoparticle tracking microscope PMX-110 (Particle Metrix, Meerbusch, Germany) was used for determining the size distribution of EVs. Camera shutter speeds of 85 and 40 were used. The instrument was calibrated with 102 nm polystyrene beads (Cat No. 900383, Microtrac, York, PA, USA).

### 2.5. Transmission Electron Microscopy (TEM)

Evs were fixed in 2.5% glutaraldehyde in 0.1 M sodium cacodylate buffer (pH 7.4). Fixed suspensions were spotted on Formvar-coated copper grids (200 mesh; Canemco, Lakefield, ON, Canada) for 30 s. Samples were negatively stained with 2% uranyl acetate in water for 6 min and dried with filter paper. Vesicles were examined on a transmission electron microscope (JEOL JEM 1230, Tokyo, Japan) that was operated at 50 kV.

### 2.6. Western Blot Analysis

Sodium dodecyl-sulfate (SDS) electrophoresis and Western blot experiments were carried out as described previously [33]. The dilution for primary antibodies was 1:1000, and the dilution for secondary antibodies was 1:5000. All antibodies were purchased from Abcam (Cambridge, UK).

### 2.7. Sample Preparation for Proteomics Analysis

The sample preparation protocol was adapted from the filter-aided sample processing (FASP) protocol [34]. Frozen EV samples in PBS, approximately 100 μL, were thawed at RT and adjusted to 0.1% *n*-Dodecyl-β-d-maltoside (DDM) (Invitrogen/Thermo Fisher Scientific, Mississauga, ON, Canada), 50 mM TRIS HCl, pH 7.5. Samples were heated to 95 °C for 3 min and cooled to RT before being transferred to a 10 kDa molecular weight cut-off (MWCO) filter (Microcon 10 K device, Cat No. MRCPRT010, Millipore, Etobicoke, ON, Canada). The buffer was exchanged by centrifuging the filter devices for 15 min at 14,000 g and adding 100 µL of denaturation buffer (8 M urea, 50 mM TRIS HCl, pH 8). Proteins were reduced by adding 4 µL of 100 mM Tris (2-carboxyethyl)phosphine (TCEP) (Thermo Fisher Scientific, Mississauga, ON, Canada) in ddH_2_O and incubation for 30 min at RT followed by centrifugation for 15 min at 14,000× *g*. 100 µL denaturation buffer and 4 µL 500 mM iodoacetamide (IAA) (Thermo Fisher Scientific, Mississauga, ON, Canada) in H_2_O were added to alkylate proteins in the dark for 45 min at RT. The reaction was then completed during the 15 min centrifugation at 14,000× *g*. Digestion buffer (50 mM TRIS-HCl, 0.6% glycerol, pH 8) was added, and the remaining urea was washed by a repeated centrifugation step before digestion buffer, and 300 ng trypsin/Lys-C (Cat No. V5071, Promega, Madison, WI, USA) were added. The digestion was maintained at 37 °C for 12 h. Peptides were collected by two centrifugation steps at 14,000× *g* for 10 min, after which 40 µL digestion buffer was added, and a 15 min centrifugation at 14,000× *g* took place. Peptides were acidified with 2 µL of 100% formic acid and desalted for MS analysis using C18 TopTip microcolumns (Cat No. TT2C18, Glygen, Washington, DC, USA) according to the manufacturer’s protocol. Purified peptides were vacuum-dried and frozen at −20 °C.

### 2.8. Surface Protein Labeling of Extracellular Vesicles

Cell culture supernatant was collected and prepared as previously described without freezing the media. Following the first ultracentrifugation steps for mEVs and sEVs at 16,500 g and 100,000 g, respectively, pellets were resuspended in 10 mM EZ-Link™ Sulfo-NHS-SS-Biotin (Cat No. 21331, ThermoFisher Scientific, Mississauga, ON, Canada) in PBS, pH 7.4. Primary amines exposed to the surface of Evs were biotinylated for 2 h at RT. The reaction was quenched by adding an equal volume of 200 mM TRIS pH 8 and incubated for 15 min. Labeled Evs were separated from free Sulfo-NHS-SS-Biotin—TRIS conjugates by diluting Evs and free linkers with PBS and an additional cleanup ultracentrifugation step as previously described. Pellets were resuspended in PBS and frozen until further processing.

### 2.9. Surface Protein Enrichment

While labeled EV samples were thawed, 120 µL (binding capacity: 60 µg of biotinylated BSA) of Pierce™ Streptavidin Agarose resin (20349, ThermoFisher Scientific, Mississauga, ON, Canada) was added on top of Pierce™ Micro-Spin Columns (#89879, ThermoFisher Scientific, Mississauga, ON, Canada) and washed three times with 1X binding buffer (0.1% DDM, 50 mM TRIS, 150 mM NaCl, 1 mM MgCl_2_, 5% glycerol, pH 7.5). Thawed EV samples were adjusted with 10X binding buffer and heated to 95 °C for 3 min. Samples were cooled down, added to the streptavidin-resin-packed columns, and labeled proteins were captured for 30 min at 4 °C while shaking. Columns were placed in new Eppendorf tubes to collect unlabeled proteins. Unbound proteins were washed five times with 1X binding buffer with 2 M Urea and collected by gentle centrifugation. Following the last wash, columns were transferred to a new tube, and 1X binding buffer with 50 mM TCEP was added to release bound proteins. The cysteine bond of the biotin linker was reduced for 2 h at RT. In parallel, the collected unbound proteins were added to a 10 kDa MWCO filter, on which the buffer was exchanged for 1X binding buffer with 50 mM TCEP and incubated for 2 h at RT. Biotinylated proteins were eluted from the streptavidin resin column by gentle centrifugation, and an additional wash with binding buffer and 2 M Urea was added on top of a 10 kDa MWCO filter for MS sample preparation.

From here, both the washed non-biotinylated and eluted biotinylated samples were treated the same. The binding buffer with TCEP was exchanged by centrifugation for 15 min at 14,000× *g* with denaturation buffer (8 M urea, 50 mM TRIS HCl, pH 8) and an additional centrifugation for 15 min at 14,000× *g*. 100 µL denaturation buffer and 4 µL 500 mM IAA in H_2_O were added to alkylate proteins in the dark for 45 min at RT. The reaction was completed during the 15 min centrifugation at 14,000× *g*. Digestion buffer (50 mM TRIS HCl, 0.6% glycerol, pH 8) was added, and the remaining urea was washed by a repeated centrifugation step before digestion buffer and 300 ng trypsin/Lys-C were added. The digestion was maintained at 37 °C for 12 h. Peptides were collected by two centrifugation steps at 14,000× *g* for 10 min, after which 40 µL digestion buffer was added, followed by 15 min of centrifugation at 14,000× *g*. The remaining peptides on top of the filter were digested by adding 100 µL digestion buffer with 10 mM CaCl_2_ and chymotrypsin (Cat No. 90056, Thermo Fisher Scientific, Mississauga, ON, Canada) for 4 h at 25 °C. Peptides were eluted as described above, acidified with 4 µL of 100% formic acid, and desalted for MS analysis using C18 TopTip microcolumns (Cat No. TT2C18, Glygen, Washington, DC, USA) according to the manufacturer’s protocol. Purified peptides were vacuum-dried and frozen at −20 °C.

### 2.10. LC-MS/MS Analysis

The LC-MS/MS analysis procedure has been described previously [8,33]. Protein samples of about 2 µg of protein, determined by Bradford assay, were injected into an UltiMate 3000 nanoRSLC (Dionex, Thermo Fisher Scientific, Mississauga, ON, Canada) coupled to an Orbitrap Fusion mass spectrometer (Thermo Fisher Scientific, Mississauga, ON, Canada).

### 2.11. MS Spectra Processing

MS raw files were analyzed with MaxQuant (version 2.0.1.0) and the Andromeda search engine [35,36]. Peptides were searched against a human UniProt FASTA file containing 20,412 entries (3 January 2019) and a default contaminants database. Default parameters were used if not mentioned otherwise. *N*-terminal acetylation and methionine oxidation were set as variable modifications, and cysteine carbamidomethylation was set as a fixed modification. A minimum peptide length of 6 amino acids was required, and the false discovery rate (FDR) was set to 0.01 for both the protein and peptide levels, determined by searching against a reverse sequence decoy database. Enzyme specificity was set as C-terminal to arginine and lysine for regular proteomics samples and including Chymotrypsin for labeled proteins with a maximum of two missed cleavages. Peptides were identified with an initial precursor mass deviation of up to 10 ppm and a fragment mass deviation of 0.5 Da. The ‘Match between runs’ algorithm in MaxQuant was performed between all samples to increase the peptide identification rate [37]. Proteins and peptides matching the reverse database were discarded. For label-free protein quantification (LFQ), a minimum ratio count of 2 was required.

### 2.12. Data Analysis

All data analysis and visualization were performed in R using the in-house-built pOmics R package (github.com/nicohuttmann/pOmics; 0.15.0; 21 September 2022). The MaxQuant output table, proteinGroups.txt, was loaded in R. Potential contaminants and reverse protein identifications, as annotated by MaxQuant, were excluded from the analysis. Proteins need to be identified by at least one unique peptide in two or more of the 12 replicate samples to be considered identified. This threshold was used to compare identified proteins by Venn diagrams between groups and the downloaded protein lists from Vesiclepedia and ExoCarta.

Principle component analysis (PCA) was performed with the *prcomp* R function based on z-scored LFQ intensity values. Proteins with a LFQ value in at least 50% of samples were used, and missing values were imputed by drawing values from a down-shifted Gaussian distribution of log-transformed protein LFQ intensities (shift = 1.8 standard deviations (sd), width = 0.3 sd) to simulate low abundance profiles [38].

Correlation matrices to predict protein co-localization were constructed for sEVs from each cell line separately by computing the pairwise Pearson correlation coefficient with R’s *cor* function from median-normalized protein LFQ intensities. Only proteins without missing values in all 12 replicates were included. Then, hierarchical clustering was computed from the resulting matrix using Euclidean distance with R’s *dist* and complete linkage clustering with R’s *hclust* function. The correlation matrices were plotted, and the position of potential biomarkers and EV proteins (CD63, CD9, and CD81) was indicated to visualize protein clusters with similar abundance profiles.

Protein subcellular localizations were retrieved with the UniProt.ws R package [39], gene ontology annotations from the org.Hs.eg.db R package [40], and gene-disease relations with the disgenet2r package [41].

## 3. Results

### 3.1. Biochemical and Morphological Characterization of Evs Isolated by Differential Ultracentrifugation

In this study, sEVs and mEVs from MDA-MB-231, MCF7, and MCF10A were isolated by differential ultracentrifugation (dUC) from cell culture supernatant as described in Material and Methods (Figure 1A and Appendix A). The isolated sEVs and mEVs were each characterized by NTA, TEM, and WB analysis. The size range of the particles was measured by NTA. The median diameter of sEVs isolated from MDA-MB-231, MCF7, and MCF10A cells was 107 nm, 110 nm, and 129 nm, while the median diameter of mEVs was 159 nm, 146 nm, and 150 nm, respectively (Figure 1B). WB analysis was used to confirm the presence of protein markers known to be present on Evs: CD63, CD9, and CD81 (Figure 1C). All EV marker proteins were present in sEV pellets from each cell line; however, CD63 was found to be less prominent in mEVs from MCF7 cells, suggesting an enrichment of Evs in the sEV pellet. TEM analysis of the isolated vesicles demonstrates the presence of an intact lipid membrane (Figure 1D). Proteomics samples were then analyzed using two approaches based on (1) global proteomic analysis and (2) enrichment of EV surface proteins by labeling surface-accessible proteins with a Sulfo-NHS-SS-Biotin reagent.

### 3.2. Proteomic Profiling of Cell Line-Derived Evs Using Global Proteomic Analysis

The proteomes of sEVs and mEVs from all three cell lines were compared. Evs from each cell line were prepared from twelve biological replicates for protein identification and quantitative analysis. A total protein content for each sample isolate of about 5 μg was analyzed by nLC-MS/MS (Figure 1A). A total of 2459 proteins were identified with at least one unique peptide in two or more replicates in mEVs and sEVs from any of the three cell lines after removing contaminant proteins (Appendix A). Proteins were mapped to gene symbols and compared with existing proteomics databases for Evs. Vesiclepedia (last updated 2018) and ExoCarta (last updated 2015) contained proteins from 13,267 to 5405 genes that were identified in Evs from human tissue or cell lines (Figure 2A), respectively [42,43]. In addition, the top 100 EV protein list, which consists of the most frequently identified proteins in EV studies, was included. All 100 proteins on this list have been identified, whereas 170 proteins have not been reported in Vesiclepedia or ExoCarta yet (Appendix A).

On average, 1483 proteins were identified per sample, with higher numbers in sEVs (Figure 2B). In sEVs and mEVs, 1315 and 1128 proteins were common to all cell lines (Figure 2C,D), and EV markers CD63, CD9, and CD81 probed by WB analysis were identified in sEVs and mEVs from each cell line. Combined, sEVs and mEVs shared 1928 proteins (Figure 2E). To compare all cell line fractions, principal component analysis (PCA) (Figure 2F) was conducted based on scaled protein abundance. Principal components 1 and 2 distinctly separate sEVs from all three cell lines, while mEVs from MCF7 and MCF10A were not separated.

### 3.3. Selection Strategy for EV BC Biomarkers

The proteomic compositions of sEVs and mEVs from three breast cell lines were analyzed to derive a list of proteins that may be further investigated for diagnostic potential for BC detection. As criteria, a filtering strategy was formulated based on differences between cell lines, protein annotations, and a prediction of which proteins were likely to co-localize with known EV markers based on protein profile correlation (Figure 3).

The observed enrichment of EV marker proteins in sEVs shifted the focus of the following analysis to proteins identified in sEVs. Initially, sEV proteins were classified into positive (only identified in MDA-MB-231, MCF7, or both sEVs) and negative markers (only identified in MCF10A sEVs), as represented in Figure 2C. Since the detected abundance should be consistent across samples containing sEVs, proteins needed to be identified by two or more peptides in all respective biological replicates, while full absence was required in sEVs and mEVs of the control cell lines. This resulted in a list of 44 proteins (Appendix A).

Next, UniProt annotations for membrane proteins were used to select proteins accessible on the surface of EVs (Appendix A). The final list of potential EV BC surface biomarkers contains 25 proteins. To predict the co-localization of the remaining 25 membrane proteins with EV marker proteins, correlation matrices were constructed for each cell line EV type (Appendix A). Pairwise Pearson’s correlation coefficients were computed from protein abundances from 12 biological replicates, and hierarchical clustering was performed on the resulting distance matrix to sort proteins into clusters of co-expression. Similar approaches have been used in the context of cell types in tissue samples, where measured protein abundances or gene counts mainly depended on the overarching composition of subgroups (cell types in tissue samples; organelles within cells; EV subtypes within isolated pellets) [44,45]. Clusters from the correlation matrices were then used to predict protein co-localization on vesicle subtypes (Appendix A) using a similar approach from recent investigations [46,47]. Finally, 3, 4, and 5 proteins were found within the same major cluster with EV marker proteins (CD63, CD9, and CD81) in each MDA-MB-231, MCF7, and MCF10A-derived sEV, respectively (Table 1). An investigative literature search of the working list of identified potential biomarkers revealed a subset of proteins with known relevance in cancer and BC that have not yet been studied in the context of EVs: Suppressor of tumorigenicity 14 protein (ST14; MDA-MB-231 and MCF7 sEV-specific), Claudin-3 (CLDN3; MCF7 sEV-specific), Tyrosine-protein kinase receptor UFO (AXL—MDA-MB-231 sEV-specific), and Integrin alpha-7 (ITGA7; MCF10A sEV-specific). The presence of AXL, ST14, CLDN3, and ITGA7 was probed by WB analysis in sEVs and mEVs derived from all three cell lines (Figure 4).

### 3.4. Western Blot Validation of AXL, ST14, CLDN3, and ITGA7 Proteins

The presence of proposed EV proteins as BC biomarkers was analyzed by WB, which also serves as a control if proposed proteins can be detected by antibodies. AXL was not detected by WB in MDA-MB-231 EVs as found by MS but was observed in MCF10A sEVs and mEVs. The presence of ST14 was detected by WB in sEVs from MDA-MB-231 and, to a lesser extent, in MCF10A, but not in MCF7. It was also fully absent in mEVs. The absence of ST14 in MCF7 sEVs contradicts the proteomics results that identified this protein in both MDA-MB-231 and MCF7 sEVs. CLDN3 was only identified in sEVs and mEVs from MCF7, which supports the proteomics data. This suggests the presence of two isoforms in sEVs and mEVs with approximate masses of 15.8 and 17.7 kDa. ITGA7 was identified only in MCF10A sEVs by proteomic analysis; notably, the full protein was found in MCF10A mEVs, and the heavy chain band with a lower mass was observed in MCF10A sEVs and faintly in MCF7 mEVs. The full WB image shows the 70 kDa form and the light chain F.

### 3.5. Proteomic Profiling of Surface EVs Using the Labeling Approach

Potential protein biomarkers on the surface of EVs were experimentally identified by the biotinylation labeling approach. Accessible lysine residues and protein *N*-termini on the surface of EVs were labeled with Sulfo-NHS-SS-Biotin, which allows for the separation of labeled and unlabeled proteins on a streptavidin affinity column and protein identification by mass spectrometry. The labeling procedure of surface proteins on EVs has been incorporated into the ultracentrifugation procedure to remove free biotin, which would otherwise interfere with surface protein enrichment. Isolated EVs were then lysed and incubated on a streptavidin-affinity column. Unlabeled proteins were washed while labeled proteins were retained and then eluted from the column by cleavage of the disulfide bond in the biotin label by TCEP. The trypsin digestion of both protein fractions was supplemented with chymotrypsin to account for missing cleavage at labeled lysine residues. Both EV pellets, sEVs and mEVs, were prepared in biological triplicate from MDA-MB-231, MCF7, and MCF10A cells.

In total, 846 proteins were found on the surface of EVs (Appendix A). A comparison with EV databases revealed 155 proteins not yet reported in EV studies (Figure 5A, Appendix A). From the top 100 EV proteins, 54 were identified on the surface of sEVs from one or more cell lines. These include EV markers such as tetraspanins (CD63, CD9, and CD81), 14-3-3 adapter proteins, and heat shock proteins. Contrary to the global proteomics approach, proteins such as histones and proteins of the Rab GTPase family were not found on the surface of vesicles.

Surface proteins identified by the labeling approach contained a high number of proteins not identified by the traditional proteomics method (Figure 5B). Many of these proteins may be underrepresented in the global proteomic analysis and, hence, are enriched as a subset of surface proteins or identified under different preparation conditions. Furthermore, the small overlap of proteins with the previous marker selection that was based on total proteomic analysis prompted an independent analysis of these results.

Based on the experimentally identified surface proteins, proteins were compared between cell lines and known disease-related proteins. With the goal of selecting BC protein markers, the focus was first on finding which proteins were common to malignant cell lines. Surface proteins of sEVs and mEVs were separately compared between cell lines, showing a large overlap between MDA-MB-231 and MCF7 sEVs (Figure 5C) and MDA-MB-231 and MCF10A mEVs (Appendix A).

Next, 88 and 32 proteins common to sEVs and mEVs from MDA-MB-231 and MCF7 that were absent in MCF10A (Appendix A), respectively, were compared with the DisGeNET database for disease associations [41]. Among the 88 and 32 identified surface proteins, 25 and 10 were annotated for several cancers. Furthermore, 9 and 2 proteins in sEVs and mEV_S_ had known implications in BC, respectively (Table 2, Appendix A). These proteins were ranked based on their intensity to approximate the abundance of proteins on the surface of EVs, which is important for diagnostic approaches. The four most abundant biotin-accessible surface proteins common to sEVs from MDA-MB-231 and MCF7 (GSTP1, EEF2, DOX10, and PGR) and two surface proteins common to mEVs from MDA-MB-231 and MCF7 (PDCD6, UTP20) may be explored for their potential as BC biomarkers in future studies. The experimental evidence of their surface accessibility increases the possibility that EVs carrying these proteins can be isolated or directly detected from plasma samples of patients.

## 4. Discussion

In this study, differential ultracentrifugation (dUC) was the method chosen to isolate sEVs and mEVs because it has been shown to provide sufficient yield and purity of EVs for proteomics analysis [48]. The originally published dUC protocols have been refined over the years by many groups with the intention of improving EV yield, sample processing time, and specificity for the EV subtype [49]. Likewise, the procedure used in previous studies of BC cell line-derived EVs [8,33] has been modified. The characterization of isolated EV pellets by NTA confirmed an overlap in the size distributions of sEVs and mEVs in all cell lines studied. This overlap has also been observed in previous investigations [8]. WB analysis identified EV markers CD9 and CD81 exclusively in sEVs of all cell lines. Additionally, while the EV marker CD63 was identified in the sEVs of all cell lines, its presence was also confirmed in the mEVs of MCF7 cells. Overall, these results indicated substantial differences between isolated sEVs and mEVs.

Following EV isolation, the protein composition was profiled by MS-based proteomics. In contrast to previous proteomics studies of BC cell line-derived EVs, the in-solution sample digestion preparation protocol [8,33] has been modified with a filter-aided sample preparation (FASP) protocol to remove interfering molecules, such as metabolites, during proteolytic digestion [34]. The observed increase in the number of proteins identified per sample in comparison to previous investigations [8,33] may be, in part, due to improved sample preparation methods. However, it is known that cell culturing conditions and chosen isolation protocols contribute to observed differences in the type and quantity of yielded protein [50]. Ultimately, the increased number of identified proteins per sample led to a higher number of commonly identified proteins between cell lines and EV fractions. Despite controlling the quantity of protein for all cell lines and fractions, sEVs from MDA-MB-231 cells gave rise to a lower number of distinct proteins per sample compared to MCF7 and MCF10A sEVs. This was previously observed in another study for triple-negative BC (TNBC) cell lines, including MDA-MB-231 [20]. Furthermore, in our previous studies [8,33], we normalized our sample preparation based on an equal number of cells. Here, normalization was based on total protein content, with each isolation sample containing 5 μg of protein. Some identified proteins (Appendix A) can be contaminants coming from the growth medium, such as albumin and growth factors. For example, growth factor C8A was found only in the MDA-MB-231 but not in the MCF7 sEVs, despite both cell lines being cultured with the same growth medium. Additionally, C8B was found in both MDA-MB-231 and MCF7 cell cultures, but in different abundances. These differences in the presence and abundance of these growth factors suggest that it is difficult to reach a conclusive answer as to whether some proteins are medium-specific contaminants or if they originated from a particular cell line.

The aim of this study was to identify EV surface proteins using two approaches for the selection of biomarkers that may have implications in BC. The first approach was based on the cell line-specific identification of sEV proteins, using UniProt annotation of membrane proteins and cluster analysis to predict co-localization with EV markers (CD63, CD9, and CD81). The analysis of protein localization by organellar maps and representation with PCA plots that has been recently applied to the study of EVs relies on fractionation of the biological sample [46]. For the organellar map, proteomic profiles from the 10,000 g, 30,000 g, and 100,000 g pellets have been combined. In the proposed approach, separate correlation matrices for sEV and mEV fractions were constructed based on the protein abundance profiles from 12 biological replicates. The 100,000 g sEV pellet is assumed to be enriched in exosomes. However, numerous studies have shown or suggested that biologically relevant EV populations overlap in their biophysical properties, such as density and size. Therefore, we reasoned that proteins within the same pellet originate from multiple biological sources, e.g., exosomes or microvesicles, and therefore show correlation patterns among proteins whose abundance changes with the abundance of their respective source. Therefore, proteins in the same clusters were assumed to be co-localized on the same EV subpopulation [44,51]. Gene ontology (GO) functional annotations and known tetraspanin marker proteins were used to indicate clusters of “core EV proteins," which were compared with the list of potential BC markers. In total, 14 proteins were predicted to be co-localized with EV markers and considered useful for further investigation.

The interaction of EV membrane proteins with other proteins that ultimately form a protein corona [52] poses a challenge for the direct detection of EV proteins or enrichment of EV-positive markers through affinity-based approaches [53]. Therefore, the first approach resulted in the identification of 11 proteins, four of which were further investigated by WB analysis. ST14, CLDN3, and ITGA7 were validated as potential BC biomarkers, while AXL was detected in MCF10A EVs instead of MDA-MB-231.

The suppressor of tumorigenicity 14 protein (ST14), also called matriptase, is a member of the transmembrane protease serine 3 (tmprss3) family [54]. ST14 has been identified by proteomics in sEVs from MDA-MB-231 and MCF7 cells, and WB confirmed its expression in MDA-MB-231 sEVs and, to a lesser extent, in MCF10A sEVs. High protein expression of ST14 has been shown in the epithelial-to-mesenchymal transition (EMT) of malignant cells isolated from human breast carcinomas [55,56,57,58]. Reduced expression of ST14 slowed tumor growth in mice by impairing the pro-HGF/c-Met signaling pathway and cell proliferation [59].

Claudins are tight junctional proteins ranging from 20 to 24 kDa in mass. They are present on the apicolateral membranes of epithelial, endothelial, and mesothelial cells [60,61]. In malignancies, the expression of claudins varies depending on the type of tumor [62]. For example, overexpression of claudins 3 and 4 has been found in several carcinomas, including BC [62]. In MCF7 cells, inhibiting CLDN3 overexpression decreased cell migration [63]. Recent findings indicate an association between the outcome of BC patients and expressions of claudins 3, 4, and 7 in membranes and cytoplasm [64]. Moreover, CLDN3 expression is positively correlated with BRCA mutations in women. Here, it was demonstrated that CLDN3 could be used as an interesting EV-based biomarker for BC.

Integrins are a major family of cell adhesion receptors. They consist of 18 alpha and 8 beta subunit genes, which can form at least 24 non-covalently linked, heterodimeric complexes from an alpha and beta subunit in mammals [65]. Integrins are well-known constituents of EVs and have been studied in different cancers [66,67]. For example, cellular ITGA7 has been found to influence migration, invasion, and epithelial–mesenchymal transition to function as a tumor suppressor gene in BC [68] and papillary thyroid carcinoma cells [69]. This study indicates that ITGA7 is highly expressed in MCF10A compared to MCF7 and MDA-MB-231-derived sEVs, which suggests that ITGA7 could be investigated as a control biomarker for BC assessments. Notably, different isoforms of ITGA7 may be explored, considering EV subtypes.

The second approach, based on surface proteomics of EVs, highlighted nine proteins with known implications in BC. These proteins were only found on the surface of EVs from MDA-MB-231 and MCF7, but not on EVs from control MCF10A cells. These include Glutathione S-transferase P1 (GSTP1), Elongation factor 2 (EEF2), DEAD/H box RNA helicase (DDX10), progesterone receptor (PGR), Ras-related C3 botulinum toxin substrate 2 (RAC2), and Disintegrin and metalloproteinase domain-containing protein 10 (ADAM10). Notably, ADAM10 has been previously identified as a potential BC biomarker [33]. Furthermore, a previous study also suggested that GSTP1, EEF2, DDX10, and PGR, which were the most abundant, might be promising biomarkers for BC detection.

A recent study demonstrated that EEF2K plays a key role in the maintenance of aggressive tumor behavior and chemoresistance in treatment-resistant TNBC [70]. In this study, EEF2K is identified in both cancerous cell lines (MDA-MB-231 and MCF7) but not in the non-cancerous control cell line (MCF10A) and may serve as a potential diagnostic biomarker. DEAD/H box RNA helicases are a group of regulating mRNA translations with growing knowledge about their involvement in cancer [71]. Reported evidence suggests that proteins of the DDX family play a crucial role in the tumorigenesis of cancer cells and stem cell differentiation [72]. Downregulation of DDX10 in BC cells led to an increased frequency of apoptosis. Therefore, the mechanism and utility of DDX10 in BC sEVs are not evident. Estrogen receptorα (ERα or ESR1) and progesterone receptorα (PGR) are crucial prognostic and predictive biomarkers in BC [73]. It has been shown that ESR1 and PGR were highly expressed in MCF7, T47D, and MDA-MB-361 metastatic cell lines, suggesting that these cancer cells represent models of estrogen- and progesterone-dependent cancers [74]. Moreover, high ESR1 and PGR expression levels were observed in ER-positive MCF7, BT-474, and T-47D BC cell lines, and low ESR1 and PGR expression was found in MDA-MB-231 [75]. These results suggest that mRNA expression levels of ESR1 and PGR can be considered distinct biomarkers and essential prognostic factors for ER-positive BC. This work identified PGR on the surface of sEVs from MCF7 and MDA-MB-231 cells and suggests that it is a potential biomarker for BC.

Two proteins from mEVs were predicted as potential surface markers of BC: programmed cell death protein 6 (PDCD6P) and small subunit processome component 20 (UTP20). PDCD6 is a pro-apoptotic protein contributing to T-cell receptor-, Fas-, and glucocorticoid-induced programmed cell death [76,77] as well as endoplasmic reticulum stress-induced apoptosis during organ formation [78,79]. PDCD6 expression was found to be upregulated in tumor tissue samples from lung, breast, colon, and ovarian cancers, which suggested that it may be involved in the maintenance of cellular viability [80,81,82]. Small subunit processome component 20 (UTP20) is present in both 90S and 40S pre-ribosome particles [83]. Furthermore, the Utp20 gene has been identified as also being associated with colorectal cancer [84].

The results obtained from both strategies resulted in a few similarities. Therefore, two separate marker selection strategies were considered. It is important to note that some EV markers, tetraspanins CD81 and CD9, were not identified in some cell line sEV samples for the biotinylated surface proteome. The method based on biotinylation requires a large sample size and is very difficult to prepare from sEVs that generally contain a low abundance of membrane proteins compared to cytosolic proteins [85]. Also, labeling by biotinylation may suffer either due to different reactivities of labeling reagents regarding lysine residues on native proteins or exposure of lysine residues due to different conformation structures and protein-protein interactions. Moreover, trypsin and chymotrypsin have different substrate specificities [86], and therefore, they can exhibit different digestion capabilities. The identification of some intracellular proteins (e.g., Zinc fingers) using biotinylation labeling could also suggest that one portion of cytosolic proteins is labeled. Overall, global proteomics and EV surface proteins by Sulfo-NHS-SS-biotin and WB analyses can have advantages or disadvantages regarding the identification of surface proteins of EVs [30].

In order to obtain more reliable BC biomarkers, future studies should investigate the presence of these candidate proteins with a larger number of cell lines and clinical samples, such as samples from healthy and BC-diseased individuals. A panel of surface protein biomarkers can further be supported by various endogenous proteins, metabolites, and RNAs (miRNA, lncRNA, and circRNA) isolated from body fluids, which were all suggested to be biomarkers for early detection and prognosis of tumors [87,88]. Therefore, all of these biomolecules from EVs could be explored for biomarker discovery in terms of the diagnosis, prognosis, and treatment of cancer.

## 5. Conclusions

The aim of this study was to identify EV surface proteins with potential as BC biomarkers. Two strategies for the selection of biomarkers were applied to derive cell line-specific EV proteins that have implications for BC. The first approach was based on using cluster analysis to predict co-localization with EV markers (CD63, CD9, and CD81) and resulted in the identification of 11 proteins, four of which were further investigated by WB analysis, revealing ST14 and CLDN3 as targets for further investigation. Surface biotinylation of EV proteins: experimentally identified proteins that are accessible from the surface of EVs. Proteins common to MDA-MB-231 and MCF7 but not identified in MCF10A were filtered for known association with BC, resulting in 9 and 2 proteins as potential surface markers of BC in sEVs and mEVs, respectively. The low number of overlapping identified biomarker proteins between two approaches suggests that multiple strategies can provide acceptable estimations of surface proteins from EVs. Finally, our data demonstrated that sEVs and mEVs represent a rich source of surface biomarkers and may be used for further exploration in carcinogenesis.

## Figures and Tables

**Figure 1 cancers-16-00520-f001:**
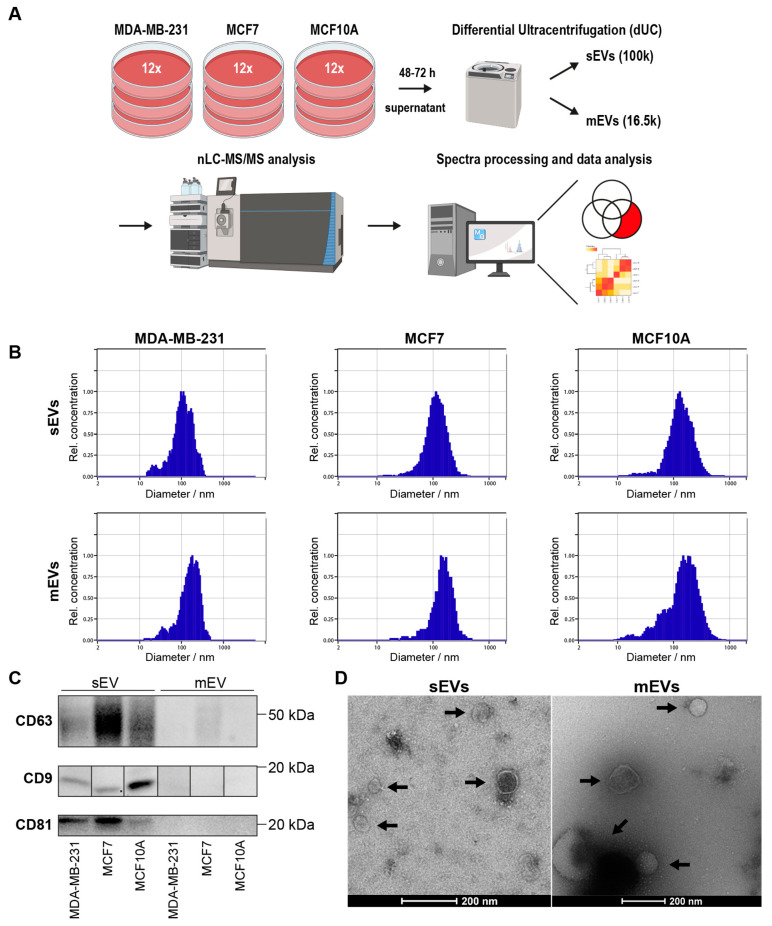
Characterization of Evs isolated from cell culture supernatant by dUC. (**A**) Workflow for EV isolation from cell culture supernatant based on dUC. (**B**) Size distribution of isolated sEVs and mEVs obtained by NTA. (**C**) WB analysis of standard EV markers CD63, CD9, and CD81 (full images in Appendix A). (**D**) TEM images of isolated sEVs and mEVs from MDA-MB-231 cells. Black arrows indicate vesicles with an intact lipid bilayer membrane.

**Figure 2 cancers-16-00520-f002:**
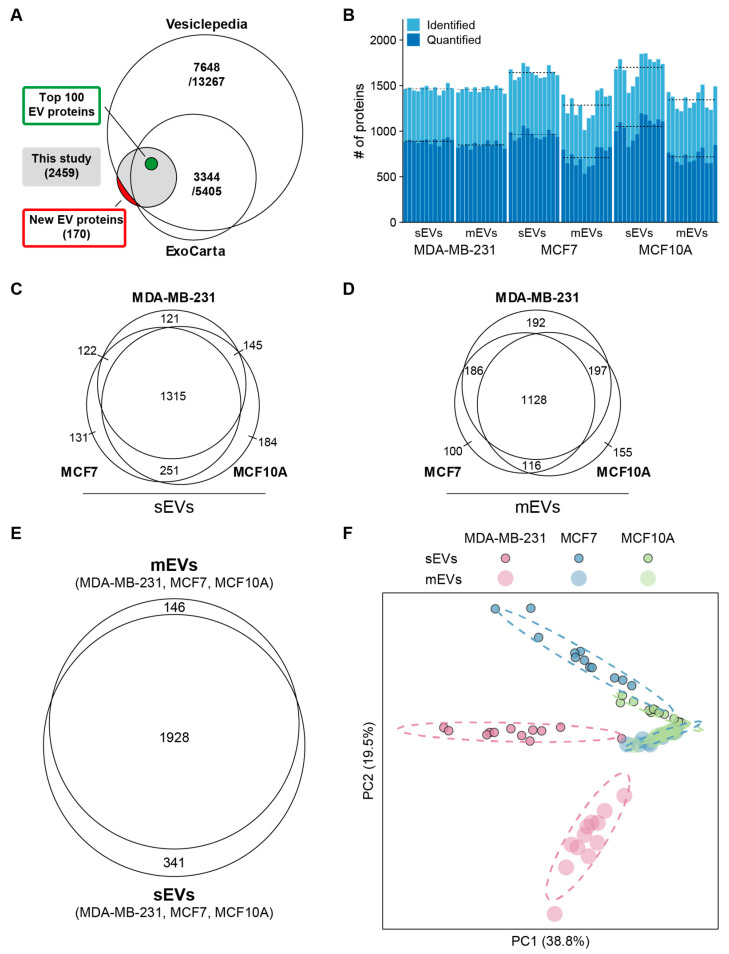
Protein profiling overview. (**A**) Venn diagram representing number of previously identified EV proteins from EV databases. (**B**) Numbers of identified and quantified proteins in each replicate. Dashed lines indicate average numbers per group. (**C**) Overlap between identified proteins in sEVs. (**D**) Overlap between identified proteins in mEVs. (**E**) Number of identified proteins in sEVs and mEVs. (**F**) PCA plot based on scaled LFQ intensities. Dashed ellipses show the 95% confidence interval for each group.

**Figure 3 cancers-16-00520-f003:**
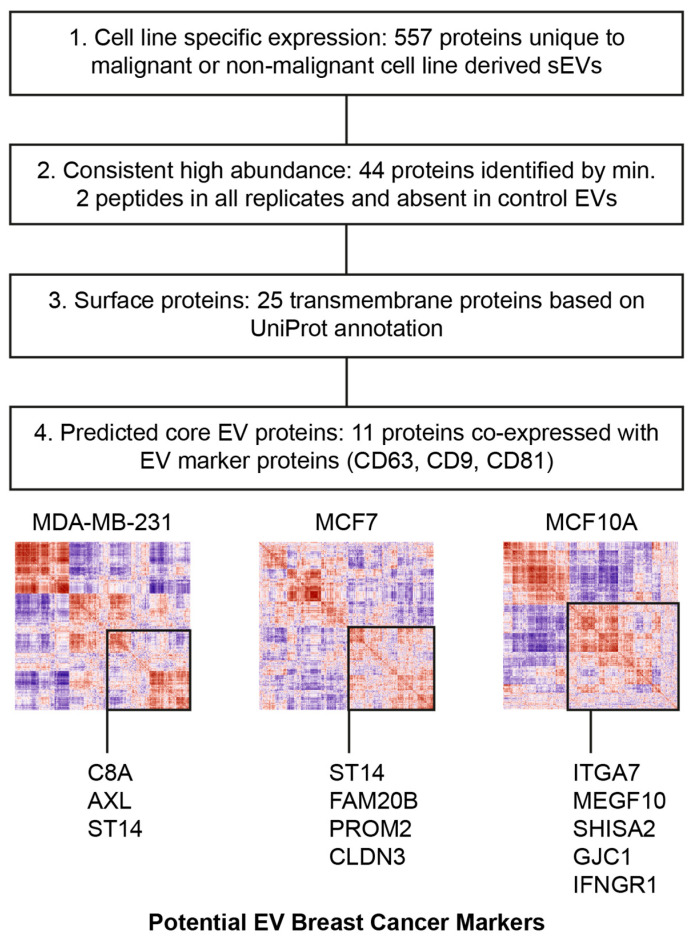
Biomarker selection strategy. 1. Proteins uniquely identified in sEVs for each cell and common to MDA-MB-231 and MCF7 were selected. 2. Abundance by at least 2 peptides in all replicates and absence of any peptides in control cell lines were required. 3. Proteins were classified by UniProt subcellular locations to identify membrane proteins. 4. Only proteins with predicted co-localization with classical EV markers based on correlation matrices were used as potential biomarkers.

**Figure 4 cancers-16-00520-f004:**
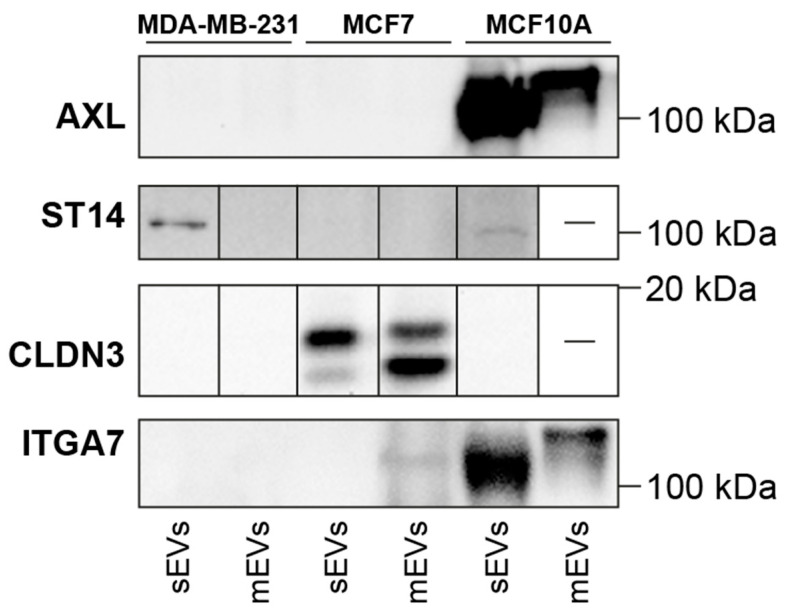
WB analysis of potential BC EV markers: AXL, ST14, CLDN3, and ITGA7 (full images in Appendix A).

**Figure 5 cancers-16-00520-f005:**
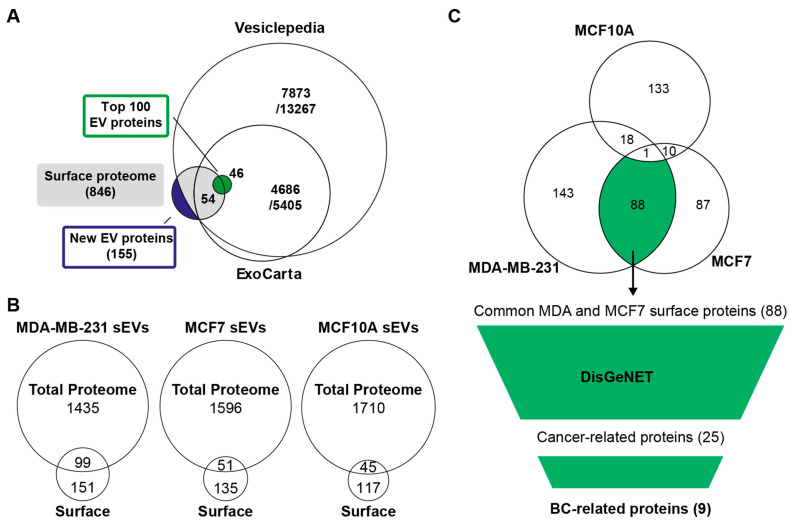
An overview of identified surface proteins on EVs from cell lines. (**A**) Comparison of all identified surface proteins from three cell lines and two EV fractions with EV databases. (**B**) Comparison of identified proteins from sEVs from each cell line from previous total proteomics analysis with surface proteins identified by surface labeling and MS analysis. (**C**) Biomarker selection based on common surface proteins from MDA-MB-231 and MCF7, not identified in MCF10A sEVs, was annotated with known disease associations from the DisGeNET database, and subsequently, proteins associated with cancer and BC were selected. The final list of potential BC EV surface markers was ranked based on LFQ intensity to prioritize more abundant proteins (Table 2).

**Table 1 cancers-16-00520-t001:** List of cell line-specific sEV transmembrane proteins proposed as biomarkers for BC.

Protein Accession	Gene Name	Protein Name
**MDA-MB-231 sEVs**		
P07357	C8A	Complement C8 alpha chain
P30530	AXL	AXL receptor tyrosine kinase
**MDA-MB-231 and MCF7 sEVs**		
Q9Y5Y6	ST14	ST14 transmembrane serine protease matriptase
**MCF7 sEVs**		
O75063	FAM20B	FAM20B glycosaminoglycan xylosylkinase
Q8N271	PROM2	Prominin 2
O15551	CLDN3	Claudin 3
Q13683	ITGA7	Integrin subunit alpha 7
**MCF10A sEVs**		
Q96KG7	MEGF10	Multiple EGF like domains 10
Q6UWI4	SHISA2	Shisa family member 2
P36383	GJC1	Gap junction protein gamma 1
P15260	IFNGR1	Interferon gamma receptor 1

**Table 2 cancers-16-00520-t002:** List of surface proteins common to MDA-MB-231 and MCF7 sEVs and mEVs with known BC associations.

Protein Accession	Gene Name	Protein Name
**sEVs**		
P09211	GSTP1	Glutathione S-transferase pi 1
P13639	EEF2	Eukaryotic translation elongation factor 2
Q13206	DDX10	DEAD-box helicase 10
P06401	PGR	Progesterone receptor
P15153	RAC2	Rac family small GTPase 2
O14672	ADAM10	ADAM metallopeptidase domain 10
Q99798	ACO2	Aconitase 2
O75691	UTP20	UTP20 small subunit processome component
Q86UW6	N4BP2	NEDD4 binding protein 2
**mEVs**		
O75340	PDCD6	Programmed cell death 6
O75691	UTP20	UTP20 small subunit processome component

## Data Availability

The raw MS data presented in this study is openly available at the PRIDE repository. This data can be found here: PXD048074.

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
