# Peer review of "Surface Proteome of Extracellular Vesicles and Correlation Analysis Reveal Breast Cancer Biomarkers"

_cancers, 2024, doi:10.3390/cancers16030520_

Round 1

Reviewer 1 Report

Comments and Suggestions for Authors

This manuscript provides detailed information on the surface proteome of extracellular vesicles and correlation analysis reveal breast cancer Biomarkers.The result supports the proposition that EVs are a rich source of known and novel biomarkers which may be used for non-invasive detection of BC. Furthermore, the presented datasets could be further explored for the identification of potential biomarkers in BC.The topic is interesting.Please revise the following comments and evaluate whether it meets the publication standards of this journal.The discussion section lacks in-depth exploration of the current research progress and significance in the relevant field. Please refer to the following literature for citation and in-depth discussion of the related content.

PMID: 36753923

PMID: 36005690

Comments on the Quality of English Language

Some grammar errors need to be corrected.

Author Response

This manuscript provides detailed information on the surface proteome of extracellular vesicles and correlation analysis reveal breast cancer Biomarkers.The result supports the proposition that EVs are a rich source of known and novel biomarkers which may be used for non-invasive detection of BC. Furthermore, the presented datasets could be further explored for the identification of potential biomarkers in BC.The topic is interesting.Please revise the following comments and evaluate whether it meets the publication standards of this journal.

We thank the reviewer for their positive and critical feedback. We agree that the state of the field regarding EVs as biomarkers can be extended.

The discussion section lacks in-depth exploration of the current research progress and significance in the relevant field. Please refer to the following literature for citation and in-depth discussion of the related content. 

PMID: 36753923 

PMID: 36005690 

In accordance with the reviewer’s comment, an additional paragraph, also outlined below, on the current state of the field and how this study may add a new perspective to ongoing research, was added in the discussion section.

Discussion:

A panel of surface protein biomarkers can further be supported by various endogenous proteins, metabolites and RNAs (miRNA, lncRNA, and circRNA) isolated from body fluids which were all suggested to be biomarkers for early detection and prognosis of tumors (PMID: 36753923; PMID: 36005690). Therefore, all of these biomolecules, from EVs, could be explored for biomarker discovery in terms of diagnosis, prognosis, and treatment of cancer. 

Reviewer 2 Report

Comments and Suggestions for Authors

To explore blood markers for breast cancers(BC), authors performed integrated extracellular vesicle analyses using two BC lines and one non-cancerous breast epithelial line. Although the result suggests the existence of potential blood markers for early diagnosis of BC, the study has limitation in that a relatively small number of cell lines, but not a large number of clinical samples, were used for the analyses. To enhance the value of the current study, an advancement of data sharing is crucial.

Major concerns:

To accelerate data sharing toward advanced BC biomarker exploration, law data should be uploaded to public database for researchers in broad fields can use them for their own purposes. 

Author Response

To explore blood markers for breast cancers(BC), authors performed integrated extracellular vesicle analyses using two BC lines and one non-cancerous breast epithelial line. Although the result suggests the existence of potential blood markers for early diagnosis of BC, the study has limitation in that a relatively small number of cell lines, but not a large number of clinical samples, were used for the analyses. To enhance the value of the current study, an advancement of data sharing is crucial. 

We thank the reviewer for their insightful comments and suggestions.

In our study, we used the MDA-MB-231 cell line with high metastatic potential and a cancerous cell line with low metastatic potential (MCF 7 is the standard for this). Both cancerous cell lines were then compared with a non-cancerous cell line, of which MCF10A is commonly used in the field.  A larger number of cell lines could provide more reliable biomarkers for BC. However, we wanted to be consistent with our previously conducted research in which we used these three cell lines: PMID: 37189694, PMID: 35203617, PMID: 33499132 and PMID: 32782317.  

Investigating the diagnostic potential of candidate biomarkers of this study in the plasma samples of breast cancer patients would require extensive collaboration and a detailed patient selection process, in addition to appropriate sample preparation methods and workflow analysis techniques. Therefore, we think that any partial and incomplete investigation may not be rigorous enough or relevant. However, in collaboration with clinical researchers, we plan to conduct a detailed clinical study in the future, to evaluate these proteins for their diagnostic and prognostic utility. 

In accordance with the reviewer’s comment, in the discussion section of our work we added:

In order to get more reliable BC biomarkers, future studies should investigate the presence of these candidate proteins with a larger number of cell lines and clinical samples such as samples from healthy and BC diseased individuals.

Major concerns: 

To accelerate data sharing toward advanced BC biomarker exploration, law data should be uploaded to public database for researchers in broad fields can use them for their own purposes.  

We thank the reviewer for their comments. We agree on the importance of data sharing and apologize that we forgot to include the reviewer account to the uploaded data.  

Please access the uploaded and described raw data under the following link https://www.ebi.ac.uk/pride/login using

Username: [email protected] and Password: OnYE6ly6. 

You can find the corresponding paragraph in the manuscript in line 596: “Data Availability Statement: The raw MS data presented in this study is openly available at the PRIDE repository. This data can be found here: PXD048074.” 

Reviewer 3 Report

Comments and Suggestions for Authors

1. Materials description is not complete:
a. sources/types of chemicals, enzymes are missing)
b. some abbreviations are not explained (eg. DDM),
c. number of replicates can be found out only from figures/results.

2. Some details of data analysis should be clarified or changed:
a. raw data on Pride repository is not publicly available, no reviewer access was given, so it could not be checked
b. as the available documentation of the applied pOmics R package is far from complete, more details of statistical analysis should be given (eg. R scripts made available)
c. it is not clear whether the 50% missing value filtration was applied on the whole dataset, or on sample groups. Were the further statistical steps (PCA, correlation) applied on the original dataset or after missing value imputation?
d. Supplementary table S1. should provide data used for correlation analysis of sEV samples, however the table also contains data for unidentified samples (V1-V36). Based on MaxLFQ intensities from this table, however I was not able to reproduce correlation clusters. Regarding the correlation analysis it is not clear what exact limit was used for “core EV” classification. Black squares on Figure 3. correspond to several major (level 2 or 3) clusters according to Fig. S4., which shows some negative between-cluster correlation coefficient too, therefore not representing similar abundance profiles. Correlation analysis in references (44,45) were applied on changes due to some perturbation, however in this work correlation is based on technical and limited biological variance of replicate cell lines. Correlation based on sEV/mEV or cell line differences (with appropriate data imputation) may give more biologically relevant clusters.
e. There is some contradiction in global and surface proteomics and WB data, which need some explanation:
- WB did not confirm the global proteomics cell-line specificity of AXL and ST14
- the data for biotinylated surface proteome is not available, but according to table S4. some EV marker tetraspanins (e.g. CD81, CD9) were not identified in some cell lines sEV samples, although having an order higher intensity in global samples then e.g. CD63, and having similar number extracellular lysines. This may be a problem with labelling or data analysis. In data analysis modification of Lys residues with remainder of labeling groups should be considered, as most of detectable chymotryptic peptides of these proteins contain Lys (although would not expect complete biotinylation). The labelled dataset contains a large number of intracellular proteins (e.g. Zinc fingers), which raises questions about surface specificity of labelling (contrary tp statements in line 385-387), however effectiveness could be statistically confirmed by enrichment analysis. The differences observed in global/surface proteome/WB are not well explained to exclude the effect of some artificial/technical bias. E.g. why NONE of the "co-localized, surface" cell-specific proteins from global analysis 
(Table 1.) were detected in the respective cells after surface enrichment (Table S4)?

f. Proteins were filtered based on Maxquant common contaminant database, however possible contaminants should be selected based on experimental setup. EVs were isolated form cell line supernatant, so contamination by growth medium can not be excluded even after centrifugation and washing steps. This is confirmed, by the presence and abundance of serum related contaminants (albumin, complement factors etc.). in every sample according to Table S1. In this work the control (MCF10A) was grown in different medium than the other cell lines, which results in significantly different level (q<0.01) of a large number of annotated contaminants (data from Table S1. which is based on intensity of shared peptides of human and bovine/horse serum proteins not included in contaminants database). The significantly different proteins include several complement members (eg. C8B), therefore C8A, which is stated to be specific to MDA (but not necessarily expressed in such cells), may be also an artifact. Therefore, I would suggest to build a medium specific contaminant library created with species specific sequence database.

3.      No quantitative comparisons were performed (except correlation), possible biomarkers were selected based on detectability compared to one cell line, but that depends on limits of detection of the sample processing-LC-MS method, and may be specific to the control cell line. The usability of the putative serum biomarkers was not estimated by detectability in serum (eg. experimentally, or based on PeptideAtlas serum database), expression levels in other organs (suppressing effects) etc., no efforts were made, to combine the results of the applied approaches, therefore it is not clear yet, how the scientific community can benefit from this work, after correction of data analysis methods.

Author Response

We thank the reviewer for their insightful comments and suggestions.

  1. Materials description is not complete: 
    sources/types of chemicals, enzymes are missing) 

For the following items description was made:

  1. 97: DMEM/F12 growth medium (Gibco™, Thermo Fisher Scientific, Mississauga, ON, Canada) 
  2. 99: fetal bovine serum (FBS) (Cat No. F2442, MilliporeSigma, Oakville, Canada) 

l.99: 100 U/mL penicillin, and 100 µg/mL streptomycin (Cat No. A5955, MilliporeSigma, Oakville, Canada) 

  1. 101: horse serum (HS) (Cat No. H1270, MilliporeSigma, Oakville, Canada) 
  2. 103: epidermal growth factor (EGF) (Cat No. GMP100-15, Peprotech, Mississauga, ON, Canada) 
  3. 104: hydrocortisone (Cat No. H-0888, MilliporeSigma, Oakville, Canada) 
  4. 105: insulin (Cat No. I-1882, MilliporeSigma, Oakville, Canada) 
  5. 106: cholera toxin (Cat No. C-8052, MilliporeSigma, Oakville, Canada) 
  6. 122: phosphate-buffered saline (PBS) (Gibco™, Thermo Fisher Scientific, Mississauga, ON, Canada), pH 7.4 
  7. 157: n-Dodecyl-β-D-maltoside (DDM) (Invitrogen/ Thermo Fisher Scientific, Mississauga, ON, Canada) 
  8. 163: Tris(2-carboxyethyl)phosphine (TCEP) (Thermo Fisher Scientific, Mississauga, ON, Canada) 
  9. 166: iodoacetamide (IAA) (Thermo Fisher Scientific, Mississauga, ON, Canada) 
  10. 168: trypsin/Lys-C (Cat No. V5071, Promega, Madison, WI, USA) 
  11. 217: chymotrypsin (Cat No. 90056, Thermo Fisher Scientific, Mississauga, ON, Canada) 

  12. some abbreviations are not explained (eg. DDM),

Following abbreviations lacking an explanation were introduced. 

  1. 122: phosphate-buffered saline (PBS)
  2. 145: Sodium dodecyl-sulfate (SDS)
  3. 152: n-Dodecyl-β-D-maltoside (DDM)
  4. 157: Tris(2-carboxyethyl)phosphine (TCEP)
  5. 166: iodoacetamide (IAA)

  6. number of replicates can be found out only from figures/results. 

In the Material and methods section, we added the number of replicates.

  1. Some details of data analysis should be clarified or changed:
    raw data on Pride repository is not publicly available, no reviewer access was given, so it could not be checked 

Please access the uploaded and described raw data under following link https://www.ebi.ac.uk/pride/login using Username: [email protected] and Password: OnYE6ly6. 

You can find the corresponding paragraph in the manuscript in line 596: “Data Availability Statement: The raw MS data presented in this study is openly available at the PRIDE repository. This data can be found here: PXD048074.” 

  1. as the available documentation of the applied pOmics R package is far from complete, more details of statistical analysis should be given (eg. R scripts made available) 

We revised section 2.12 Data Analysis to describe more details related to all aspects as mentioned below.

  1. c. it is not clear whether the 50% missing value filtration was applied on the whole dataset, or on sample groups. Were the further statistical steps (PCA, correlation) applied on the original dataset or after missing value imputation?

To clarify all questions related to the analysis of the data, we revised section 2.12. Data Analysis in the Materials and methods. It now contains precise treatment of the data for each analysis step and the necessary R functions.

  1. Supplementary table S1. should provide data used for correlation analysis of sEV samples, however the table also contains data for unidentified samples (V1-V36). Based on MaxLFQ intensities from this table, however I was not able to reproduce correlation clusters.

We added three supplemental tables (S4-S6) which contain the ordered values of the correlation matrices with color coding allowing exploration of the clusters with and without coding experience.

Table S4 Source data for correlation matrix of MDA-MB-231 sEVs (Figure S4). Only proteins with complete LFQ intensities in all 12 replicates were considered. Pairwise Pearson correlation coefficients were sorted based on Euclidean distance and complete-linkage clustering.

Table S5 Source data for correlation matrix of MCF7 sEVs (Figure S4). Only proteins with complete LFQ intensities in all 12 replicates were considered. Pairwise Pearson correlation coefficients were sorted based on Euclidean distance and complete-linkage clustering.

Table S6 Source data for correlation matrix of MCF10A sEVs (Figure S4). Only proteins with complete LFQ intensities in all 12 replicates were considered. Pairwise Pearson correlation coefficients were sorted based on Euclidean distance and complete-linkage clustering.

In the Methods section we revised a paragraph in section 2.12. Data Analysis:

“Correlation matrices to predict protein co-localization were constructed for sEVs from each cell line separately by computing the pairwise Pearson correlation coefficient with R’s cor function from median-normalized protein LFQ intensities. Only proteins without missing values in all 12 replicates were included. Then, hierarchical clustering was computed from the resulting matrix using Euclidean distance with R’s dist function and complete linkage clustering with R’s hclust function. The correlation matrices were plotted, and the position of potential biomarkers and EV proteins (CD63, CD9, and CD81) was indicated to visualize protein clusters of similar abundance profiles.”

The samples V1-V36 correspond to MV samples, and a meta table uploaded to the PRIDE repository explains all sample names.

Regarding the correlation analysis it is not clear what exact limit was used for “core EV” classification. Black squares on Figure 3. correspond to several major (level 2 or 3) clusters according to Fig. S4., which shows some negative between-cluster correlation coefficient too, therefore not representing similar abundance profiles. Correlation analysis in references (44,45) were applied on changes due to some perturbation, however in this work correlation is based on technical and limited biological variance of replicate cell lines. Correlation based on sEV/mEV or cell line differences (with appropriate data imputation) may give more biologically relevant clusters.

Correct, there is negative intra-cluster correlation. This is biologically plausible and is considered by more sophisticated network analysis concepts such as the topological overlap measure. We argue that proteins can correlate with many other proteins and belong to the same subpopulation of vesicles but show an individual negative correlation. As we did not apply further network analysis schemes, we relied on manual analysis of the correlation clusters. 

Other correlation analysis studies do focus on perturbations, which in the case of extracellular vesicles has been a long-standing challenge. The 100,000g pellet, that we refer to as small extracellular vesicles, are assumed to be enriched in exosomes. However, numerous studies have shown and suggested that biologically relevant EV subpopulations, within e.g. exosomes, may not be separable by their biophysical properties. Therefore, we reasoned that a co-expression of the respective proteins of each subpopulation may result in subtle covariance which we used for our analysis strategy. 

In the discussion section we added:

The 100,000g sEV pellet is assumed to be enriched in exosomes. However, numerous studies have shown and suggested that biologically relevant EV populations overlap in their biophysical properties such as density and size. Therefore, we reasoned that proteins within the same pellet originate from multiple biological sources, e.g. exosomes or microvesicles, and therefore show correlation patterns among proteins whose abundance changes with the abundance of their respective source.

  1. There is some contradiction in global and surface proteomics and WB data, which need some explanation:
    - WB did not confirm the global proteomics cell-line specificity of AXL and ST14
    - the data for biotinylated surface proteome is not available, but according to table S4. some EV marker tetraspanins (e.g. CD81, CD9) were not identified in some cell lines sEV samples, although having an order higher intensity in global samples then e.g. CD63, and having similar number extracellular lysines. This may be a problem with labelling or data analysis. In data analysis modification of Lys residues with remainder of labeling groups should be considered, as most of detectable chymotryptic peptides of these proteins contain Lys (although would not expect complete biotinylation). The labelled dataset contains a large number of intracellular proteins (e.g. Zinc fingers), which raises questions about surface specificity of labelling (contrary tp statements in line 385-387), however effectiveness could be statistically confirmed by enrichment analysis. The differences observed in global/surface proteome/WB are not well explained to exclude the effect of some artificial/technical bias. E.g. why NONE of the "co-localized, surface" cell-specific proteins from global analysis (Table 1.) were detected in the respective cells after surface enrichment (Table S4)? 

In accordance with the reviewer’s suggestion in the results section we added:

From the Top 100 EV proteins, 54 were identified on the surface of sEVs from one or more cell lines. These include EV markers such as tetraspanins (CD63, CD9, CD81), 14-3-3 adapter proteins, and heat shock proteins.

We already discussed the results of the WB analysis in contradiction with global proteomics. However, it is true that we did not sufficiently discuss the overall results regarding global/surface proteomics and WB data. In accordance with the reviewer’s comments, an additional paragraph was added in the section of discussion: 

It is important to note that some EV markers, tetraspanins, CD81 and CD9, were not identified in some cell line sEV samples for the biotinylated surface proteome. The method based on biotinylation requires a large sample size and is very difficult to prepare from sEVs that in general contain low abundance of membrane proteins compared to cytosolic proteins size [85]. Also, labeling by biotinylation may suffer either due to different reactivities of labeling reagents regarding lysine residues on native proteins or exposure of lysine residues due to different conformation structure and protein-protein interactions. Moreover, trypsin and chymotrypsin have different substrate specificities [86] and therefore they can exhibit different digestion capabilities. The identification of some intracellular proteins (e.g. Zinc fingers) using biotinylation labeling also could suggest that one portion of cytosolic proteins is labelled. Overall, global proteomics and EV surface proteins by Sulfo-NHS-SS-biotin and WB analyses can have advantages or disadvantages regarding the identification of surface proteins of EVs.

  1. Proteins were filtered based on Maxquant common contaminant database, however possible contaminants should be selected based on experimental setup. EVs were isolated form cell line supernatant, so contamination by growth medium can not be excluded even after centrifugation and washing steps. This is confirmed, by the presence and abundance of serum related contaminants (albumin, complement factors etc.). in every sample according to Table S1. In this work the control (MCF10A) was grown in different medium than the other cell lines, which results in significantly different level (q<0.01) of a large number of annotated contaminants (data from Table S1. which is based on intensity of shared peptides of human and bovine/horse serum proteins not included in contaminants database). The significantly different proteins include several complement members (eg. C8B), therefore C8A, which is stated to be specific to MDA (but not necessarily expressed in such cells), may be also an artifact. Therefore, I would suggest to build a medium specific contaminant library created with species specific sequence database.  

According to the reviewer’s comment in the discussion

results section of our work we added.

Some identified proteins (Table 1S) can be contaminants coming from the growth medium such as albumin and growth factors. For example, growth factor C8A was found only in the MDA-MB-231 but not in the MCF7 sEVs despite both cell lines were cultured with the same growth medium. Additionally, C8B was found in both MDA-MB-231 and MCF7 cell cultures but with different abundances. These differences in the presence and abundance of these growth factors suggest that it is difficult to reach a conclusive answer whether some proteins are medium-specific contaminants or if they originated from a particular cell line.

3.      No quantitative comparisons were performed (except correlation), possible biomarkers were selected based on detectability compared to one cell line, but that depends on limits of detection of the sample processing-LC-MS method, and may be specific to the control cell line. The usability of the putative serum biomarkers was not estimated by detectability in serum (eg. experimentally, or based on PeptideAtlas serum database), expression levels in other organs (suppressing effects) etc., no efforts were made, to combine the of the applied approaches, therefore it is not clear yet, how the scientific community can benefit from this work, after correction of data analysis methods.

To investigate the diagnostic potential of candidate biomarkers identified in this study, it is necessary to compare these proteins with the proteomics analysis of EVs from plasma samples of breast cancer patients. Currently the proteome of EVs of breast cancer patients is not available. Therefore, future study should focus on the sample preparation methods for the isolation of EVs from plasma samples and its proteomics analysis. Furthermore, the EV proteomics data may show new and useful signatures for proposed biomarkers compared to protein levels from tissues or blood which were not enriched for EVs.

Reviewer 4 Report

Comments and Suggestions for Authors

The manuscript is based on research from the scientific literature and presents information on using biomarkers as a less invasive method of breast cancer (BC) diagnosis. This study supports the proposition that extracellular vesicles (EVs) are a rich source of biomarkers that may be used for the non-invasive detection of BC. Furthermore, the presented datasets could be further explored for the identification of potential biomarkers in BC.

The studies were performed on two breast cancer cell lines, MDA-MB-231 and MCF7, and on a non-cancerous breast epithelial cell line MCF10A.

In the Western Blotting experiments, was beta-actin added as a positive control?

In general, the quality of the article is good and, overall, the manuscript is interesting to readers. English language and style are good, but there are some minor spelling mistakes. In conclusion, I consider the article could be a useful contribution to the journal. I recommend the manuscript be published.

Author Response

The manuscript is based on research from the scientific literature and presents information on using biomarkers as a less invasive method of breast cancer (BC) diagnosis. This study supports the proposition that extracellular vesicles (EVs) are a rich source of biomarkers that may be used for the non-invasive detection of BC. Furthermore, the presented datasets could be further explored for the identification of potential biomarkers in BC. 

The studies were performed on two breast cancer cell lines, MDA-MB-231 and MCF7, and on a non-cancerous breast epithelial cell line MCF10A. 

In the Western Blotting experiments, was beta-actin added as a positive control? 

In general, the quality of the article is good and, overall, the manuscript is interesting to readers. English language and style are good, but there are some minor spelling mistakes. In conclusion, I consider the article could be a useful contribution to the journal. I recommend the manuscript be published. 

We thank the reviewer for their positive comments on the manuscript.

Although beta-actin has sometimes been identified as an integral part of cellular EVs, the scientific community in the field of EVs does not recommend its use as a positive control in Western Blotting experiments. We reasoned that the presence of the tetraspanin markers CD63, CD9 and CD81 would suffice as a qualitative loading control, but indeed we did not use a quantitative loading control such as beta-actin.  In the field of EV research, it is really a challenge to find a quantitative marker, but this requires a special study and the use of a large number of cell samples in order to prove such a possibility.

Round 2

Reviewer 1 Report

Comments and Suggestions for Authors

None

Comments on the Quality of English Language

none

Reviewer 2 Report

Comments and Suggestions for Authors

Authors have provided a proper explanation to resolve the concerns. The current manuscript is a strong candidate for publication in Cancers.

Reviewer 3 Report

Comments and Suggestions for Authors

There are still concerns about correlations and surface enrichement results, but readers are informed about those in the added paragraphs. Otherwise good quality data and representation.